# Self-Reported COVID-19 Vaccine Hesitancy and Willingness to Pay: A Cross-Sectional Survey in Thailand

**DOI:** 10.3390/vaccines10040627

**Published:** 2022-04-16

**Authors:** Kulpatsorn Mueangpoon, Chapipak Inchan, Panithan Kaewmuneechoke, Peerunda Rattana, Supanut Budsratid, Suthasinee Japakiya, Pitchayanont Ngamchaliew, Polathep Vichitkunakorn

**Affiliations:** 1Faculty of Medicine, Prince of Songkhla University, Songkhla 90110, Thailand; 5910310140@psu.ac.th (K.M.); 6010310019@psu.ac.th (C.I.); 6010310051@psu.ac.th (P.K.); 6010310069@psu.ac.th (P.R.); 6010310153@psu.ac.th (S.B.); 6010310159@psu.ac.th (S.J.); 2Department of Family and Preventive Medicine, Faculty of Medicine, Prince of Songkhla University, Songkhla 90110, Thailand; pitchayanont@hotmail.com

**Keywords:** COVID-19, vaccine hesitancy, willingness to pay, vaccine accessibility, vaccine comprehension

## Abstract

This study aimed to estimate the prevalence and influencing factors of COVID-19 vaccine hesitancy and willingness to pay in Thailand. A descriptive cross-sectional study was conducted from 13 September 2021 to 14 January 2022. Data were collected using an online questionnaire consisting of demographic characteristics, COVID-19 vaccine hesitancy (delay in acceptance and denying vaccination), determinants of vaccine hesitancy (complacency, convenience, and confidence), and willingness to pay. The general Thai population aged ≥18 years were surveyed. Among 705 respondents, 10.4% reported hesitancy, with significant determinants being low complacency and confidence in the vaccine; low convenience was not a significant determinant. Multivariate analysis revealed vaccine hesitancy among women, those with higher education, non-healthcare workers, and those who lived in rural areas. Furthermore, 77.2% of respondents were willing to pay, with the majority willing to pay in the range of THB 501–1000 ( USD 1 = THB 33) per dose. Increased monthly income, no impact of COVID-19 on income, and time period (before mRNA vaccine availability) significantly affected willingness to pay.

## 1. Introduction

COVID-19 is caused by the SARS-CoV-2 virus and can lead to severe respiratory symptoms [1]. Previous evidence has shown that fully vaccinated people have a lower incidence rate of COVID-19 infection, COVID-19-related hospitalization, severe symptoms, and a lower mortality rate than unvaccinated people [2,3]. Therefore, mass vaccination has become one of the most effective ways to decrease the burden of COVID-19. However, vaccine availability, including type, varies between countries.

The World Health Organization (WHO) defines vaccine hesitancy as a “delay in acceptance or refusal of vaccine despite the availability of vaccine services” and divides the determinants behind such hesitancy into three components (3C): complacency, convenience, and confidence [4] and five components (5C): confidence, complacency, calculation of risk, and collective responsibility [5]. A study in Italy found that the most common reason people refused vaccination was a lack of information on the benefits and safety of vaccines [6]. Additionally, socio-demographic factors affected vaccine hesitancy [7]. Surveys conducted in several countries have shown that many people are prone to vaccine hesitancy [7,8,9]; therefore, these issues must be explored to encourage vaccine acceptance.

Due to the novelty of the globally used COVID-19 vaccines, policymakers require information on people’s willingness to pay (WTP) and acceptable pricing to increase vaccine acceptance through policy. Unwillingness to pay was associated with a belief that the government is responsible for providing free vaccination [10].

In Thailand, the government and private sector allocate vaccines. Initially, the government relied solely on Sinovac and AstraZeneca, which were acquired in February and June 2021, respectively. The government’s free vaccine program was provided to healthcare workers and patients with one of seven underlying diseases in the first half of 2021 and was subsequently offered free to the public in the late third quarter of 2021. In addition, Thailand received a donation of one million doses of the Pfizer mRNA-based vaccine from the U.S., which arrived in August 2021, to address public demand for mRNA-based vaccines. However, due to their limited number, the mRNA-based vaccines were not offered until December 2021. During that time, the government allowed the private sector and other organizations, such as the Government Pharmaceutical Organization, to import Sinopharm, which was acquired in June 2021, and the Moderna mRNA-based vaccine, which arrived in November 2021. Vaccines offered by the private sector were administered at a set price per dose and were not subsidized by the government. Fully paid reservations were required for mRNA-based vaccines at private hospitals. However, beginning in November 2021, all vaccines became available to the general population free of charge. Therefore, it is essential to explore the factors influencing WTP that vary over time.

The rationale of this study was that few studies investigated COVID-19 vaccine hesitancy and WTP in low- and middle-income countries, especially in Southeast Asia. However, the prevalence of vaccine hesitancy in 32 countries varied from 2–62% [11,12] due to differences in methodologies (i.e., questionnaire and survey time frame), cultures, and government policy. From an economic view, many low- and middle-income countries might not be able to distribute the free vaccines to the entire population. Moreover, there were limited existing studies in health economics on the COVID-19 vaccine in these countries [13,14,15,16]. Following these gaps in knowledge, this study sought to provide information to promote vaccination policies and vaccine perspectives. To reduce vaccine hesitancy, policies should be tailored to cultures and be more specific to target populations, for example, encouraging vaccine confidence in rural areas and improving concern regarding side effects in non-healthcare providers. For WTP, our policymakers can design a public-community partnership model for COVID-19 accessibility and forecast the government’s budget for vaccination of COVID-19 or other emerging infectious diseases in the future. This study aimed to estimate the prevalence and influencing factors of vaccine hesitancy and WTP for the COVID-19 vaccine in the general Thai population from late 2021 to early 2022. We hypothesized that the prevalence of vaccine hesitancy in Thailand was less hesitant and more WTP to receive the high-efficacy vaccine if participants perceived that a vaccine with an affordable price could protect them from severe COVID-19 infection. Additionally, vaccine hesitancy might be associated with WTP. 

## 2. Materials and Methods

### 2.1. Study Design and Setting

This cross-sectional study was conducted among the general Thai population from 13 September 2021 to 14 January 2022. Online questionnaires were distributed through the Facebook and Instagram social media platforms. We also placed posters with a QR code for recruiting participants in public places. Each participant read the invitation message and obtained an e-consent before answering the questionnaire. They could withdraw from the study at any time. We did not provide any monetary or other compensation to our participants. Our research protocol was approved by the Human Research Ethics Committee of the Faculty of Medicine, Prince of Songkhla University (REC.64-365-9-1). Written informed consent was waived by the e-consent committee. 

### 2.2. Study Sample, Sampling, and Sample Size Calculation

The respondents were vaccinated (i.e., suddenly vaccinated and delayed vaccinated) and unvaccinated (i.e., denied vaccination and those who have not been offered any vaccines) Thai people aged ≥18 years who owned smartphones or could access the internet services. The exclusion criteria were cognitive impairments. For the sampling technique, we randomly recruited all participants who had completed the survey to calculate the sample size for estimating an infinite population proportion or prevalence of vaccine hesitancy and WTP. 

The sample size was calculated by using data from a previous study by Wouters et al. [11]. The proportion of COVID-19 vaccine hesitancy in the Japanese population was 0.33. The margin of error was 0.04 (10% to 15% of the proportion). Thus, a minimum of 531 participants was required in this study.

### 2.3. Data Collection 

Data were collected using an online, anonymous, self-administered questionnaire designed by KoBoToolbox (humanitarian technology, Cambridge, USA). We did not collect any personally identifiable data. The questionnaire comprised three parts: demographic information (seven questions), vaccine hesitancy (11 questions), and WTP (two questions) for the vaccine. The questionnaire was developed based on the Oxford COVID-19 vaccine hesitancy scale [17], the definition provided by the WHO [8], and the WTP scale [18,19]. The Oxford COVID-19 vaccine hesitancy scale scores correlated with the scores on the vaccine hesitancy scale (correlation coefficient = 0.47, *p* < 0.001) [20]. The Cronbach’s alpha was 0.97 [17]. Each item was validated by three content experts using face validity before a pilot study was conducted with 30 respondents to ensure a similar understanding.

### 2.4. Measure

#### 2.4.1. Independent Variables

The independent variables were gender, age, marital status, religion, education, healthcare workers, and residence. Based on the United Nations (U.N.) guidelines on standard international age classification [21], we categorized our participants into four groups: between 18 and 24 years, between 25 and 44 years, between 45 and 64 years, and over 65 years. Residence was classified into three types: rural, suburban, and urban. The healthcare worker variable was collected using the question, “Do you work in a healthcare settings?” (answer: Yes/No). The impact of COVID-19 on income was collected using the question, “Did the COVID-19 pandemic affect your income?” (answer: Yes/No). The duration was divided into two periods: before 1 November 2021 and after 1 November 2021. This date was set according to the availability of the Moderna mRNA-based vaccine that was available for a fee starting 1 November 2021 and was provided by the private sector, as well as the availability of Pfizer and Moderna mRNA-based vaccines that were made available by the government free of charge beginning at the end of November 2021.

#### 2.4.2. Dependent Variable—COVID-19 Vaccine Hesitancy and Its Determinants

Vaccine hesitancy is a complex and context-specific term that varies across time, place, and vaccine. This study defined delay in acceptance as having received two doses or more at least two weeks from the last dose and previously refusing vaccination. Denying vaccination was defined as refusing to get vaccinated up until the time of the study.

The determinants of vaccine hesitancy were complacency, convenience, and confidence. Complacency was defined as the risk of COVID-19 infection, the benefit of the vaccine, impact on daily life, and support for vaccination. Convenience was defined as vaccine accessibility, including the ability to pay, physical accessibility, geographical accessibility, and process comprehension, including sufficient language knowledge to understand the information, and vaccine comprehension, including perception and knowledge regarding the vaccine. Confidence was defined as trust in vaccine effectiveness, lack of concern regarding side effects, trust in vaccine delivery policies, including the reliability and competence of health services and healthcare professionals, and vaccine necessity.

Respondents were asked about their refusal of vaccination in the past, unwillingness to receive a vaccination, and predictive factors of hesitancy in three domains: competency, convenience, and confidence (Table 1). Answers were scored on a five-point Likert scale (1 = strongly disagree, 2 = disagree, 3 = uncertain, 4 = agree, 5 = strongly agree). The average score was calculated using the median with interquartile range (IQR) for each question.

#### 2.4.3. Dependent Variable-WTP

WTP is a measure of the maximum amount of money that consumers are willing to pay to obtain a vaccine of a given quality to increase health benefits [14,16]. In this study, respondents were asked about their WTP for the COVID-19 vaccine. If respondents were willing to pay, the following question was asked: “Rather than vaccines provided by government, how much are you willing to pay for alternative COVID-19 vaccines (THB per dose)?” The range of WTP was classified into seven scales: 125–250, 251–500, 501–1000, 1001–2000, 2001–4000, 4001–8000, and over 8000 THB (USD 1 = THB 33). Furthermore, respondents who were not willing to pay for the vaccine were asked if they were willing to pay any amount.

### 2.5. Statistical Analysis 

The data collected from the respondents were entered into a team-shared Microsoft Excel spreadsheet and analyzed using R software version 4.1.0 [22]. For data management, we applied a complete-case analysis for each outcome variable (i.e., vaccine hesitancy and WTP). Descriptive statistics, such as percentage (%), median with interquartile range (IQR), and median with minimum and maximum, were used to demonstrate the respondents’ demographic characteristics and vaccination history. Comparisons between our independent variables and vaccine hesitancy or WTP were performed using a Chi-squared test or Fisher’s exact test for categorical data and the Mann–Whitney U test for continuous data. A heatmap or highlight table was used to visualize the relationship between WTP and demographic data and vaccine hesitancy. The color scale was green to yellow and then red, with a high percentage colored green and a low percentage colored red. Binary logistic regression models were applied to measure the magnitude of the independent variables that affected vaccine hesitancy and WTP, with an odds ratio (OR) and a 95% confidence interval (CI). A univariate analysis was first performed to identify any potential independent variables. For multivariable analysis, potential independent variables with a *p*-value < 0.2 according to a univariate analysis were included in the initial model. A manual backward stepwise refinement was performed for the final model. The refined model had to pass the multicollinearity assumption. A variance inflation factor (VIF) greater than five indicated collinearity between our independent variables. Statistical significance was set at *p*-value < 0.05.

## 3. Results

Among the 705 respondents, 585 (83.0%) participants (i.e., vaccinated, delayed vaccinated, and denied vaccinated) provided answers regarding vaccine hesitancy. For WTP, 650 (92.2%) participants provided answers (i.e., vaccinated, delayed vaccinated, denied vaccinated, and those who had not been offered any vaccines).

### 3.1. COVID-19 Vaccine Hesitancy and Its Influencing Factors

The 705 respondents were predominantly female, single, and Buddhist, had a bachelor’s degree or above, and were living in an urban area (Table 2). A total of 10.4% of the respondents demonstrated hesitancy. Most of the respondents were in the delayed-vaccinated group. There were statistically significant differences by sex, age, marital status, religion, education level, being a healthcare worker, residence, and monthly income (*p* < 0.001). The proportion of hesitancy in women and the age group between 18 and 24 years was nearly twice that in men and those between 25 and 44 years of age, respectively.

Multivariate logistic regression showed that vaccine hesitancy was significantly higher in women (adjusted OR [AOR] 1.11; 95%CI 1.01–1.23), those holding a bachelor’s degree or above (AOR 1.28; 95%CI 1.04–1.57), non-healthcare workers (AOR 1.23; 95%CI 1.17–1.29), those living in rural areas (AOR 1.22; 95%CI 1.09–1.05), and those with a monthly income between THB 20,001 and 40,000 (AOR: 1.15; 95%CI: 1.10–1.21) (Table 3). Hesitancy was significantly lower in the age group between 25 and 44 years than in that between 18 and 24 years (AOR 0.89; 95%CI 0.86–0.93).

### 3.2. Determinants of COVID-19 Vaccine Hesitancy

The determinants of vaccine hesitancy were divided into three domains: complacency, convenience, and confidence. Regarding complacency, the perception of a COVID-19 infection risk was not significantly different between the hesitant and non-hesitant respondents based on a comparison of median scores (3.8 vs. 3.9, *p* = 0.460, Table 4); however, the beneficence of the vaccine (4.2 vs. 4.5, *p* = 0.009), daily life impact (3.3 vs. 3.7, *p* = 0.001), and support for vaccination (4.1 vs. 4.5, *p* < 0.001) were significantly different. In terms of confidence, the results were significantly different for lack of concern regarding side effects (2.7 vs. 3.2, *p* < 0.001), trust in effectiveness (2.6 vs. 3.2, *p* < 0.001), trust in vaccine policies (2.6 vs. 3.1, *p* < 0.001), and vaccine necessity (3.8 vs. 4.4, *p* < 0.001). In regard to convenience, the results for vaccine accessibility, process comprehension, and vaccine comprehension were not significantly different between hesitant and non-hesitant respondents (*p* > 0.05).

### 3.3. WTP and Influencing Factors for the COVID-19 Vaccine

Among the 705 respondents, 650 (92.2%) provided answers regarding WTP. Over half of the respondents were female, were single, had completed higher education, were living in an urban area, were vaccinated after 1 November 2021, were non-hesitant in regard to getting vaccinated against COVID-19, and had experienced an impact from COVID-19 on their income (68.3%, 58.6%, 77.2%, 65.5%, 75.5%, 62.1%, and 88.3%, respectively).

The study revealed that three-fourths of respondents were willing to pay for the COVID-19 vaccine (Table 5). Univariate analysis showed significant differences in marital status, age group, education, monthly income, the impact of COVID-19 on income, duration of vaccination, and vaccine hesitancy.

Multivariate logistic regression, presented in Table 6, demonstrated that the WTP was significantly higher in respondents who did not experience an impact on their income (AOR 1.31; 95%CI 1.06–1.68), while it was significantly lower in respondents who received the vaccine after 1 November 2021 (AOR 0.90; 95%CI 0.84–0.97). Although the WTP proportion increased based on earnings, only respondents with an income ranging between THB 10,000–20,000 and those with income above THB 40,000 willingly chose to pay for the vaccine (AOR 1.05; 95%CI 1.02–1.07 and AOR 1.67; 95%CI 1.44–1.93, respectively).

Interestingly, while 11.7% of the respondents were hesitant regarding COVID-19 vaccination, 83.6% were hesitant to pay for the vaccine, which was significantly different based on the univariate analysis (*p* < 0.001). However, no significant difference was detected between the hesitant and non-hesitant groups in the multivariate logistic regression analysis (AOR 1.01; 95%CI 0.99–1.04).

### 3.4. Range of WTP for COVID-19 Vaccines

Most of the 137 (27.3%) respondents were willing to pay for a vaccine in the range of THB 501–1000, whereas some groups of respondents were willing to pay in the range of THB 1001–2000, including single people (26.6%), those aged between 25 and 44 years (30.1%), those who received vaccination after 1 November 2021 (27.5%), and those who exhibited vaccine hesitancy (25.5%). Meanwhile, only 6.1% of residents in rural areas were willing to pay in the range of THB 4001–8000, and 59% of people earning a monthly income lower than THB 5000 were willing to pay in the range of THB 125–250. Furthermore, respondents were rarely willing to pay more than THB 4000 per dose (Table 7).

### 3.5. Reasons for Unwillingness to Pay for Vaccines

Reasons for not being willing to pay for the vaccine included that the supply of government vaccines was insufficient (29%), the government should pay for vaccines (25%), they do not have enough money to cover the vaccine cost (20%), people who spread the virus should pay for vaccines (10%), vaccination is immoral (10%), and other tasks taking priority over vaccination (6%).

## 4. Discussion

### 4.1. Principal Findings and Previous Studies 

In 2021, COVID-19 vaccination has increased in several countries to increase mass immunization. Mixed and combined COVID-19 vaccine regimens, for instance, two doses of a viral vector vaccine (AstraZeneca) or whole-pathogen inactivated vaccine (Sinovac) followed by booster doses of mRNA-based vaccines (Pfizer or Moderna), are generally used in Thailand. Uncertainty in relation to vaccine novelty with the unapproved efficacy of mixed regimens and concerns regarding side effects may challenge the vaccine acceptance rate among hesitant people globally.

While small groups of people felt hesitant toward COVID-19 vaccination, the overall direction of the results indicated that there were some points related to hesitancy. Being a woman, having graduated with a bachelor’s degree or above, being a non-healthcare worker, and living in a rural area were significant independent predictors of hesitancy. 

This study found that 10.4% of all respondents reported being hesitant, which was less than shown in previous studies in Singapore (9.9%) [23], Australia (15–40%) [24,25], Brazil (17.5%) [26], China (18–23%), Italy (25–31%) [6,27,28], the U.S. (29–33%) [29,30], Iraq (38.2%) [31], and Lebanon (78.6%) [32]. It appears that the number of hesitant people was lower than in other studies, which may be due to cultural barriers or the failure of government policy to convince people to receive the vaccines. Based on these results, encouraging people to receive the vaccines may be more easily achievable through government policy and may be more practical in countries with less hesitant populations. Although Thailand has a lower ratio of hesitant individuals, these individuals should not be overlooked because they may disrupt the development of herd immunity.

The results clearly demonstrated that complacency and confidence, particularly concerning side effects and trust in the effectiveness and policy, were significantly different between the hesitant and non-hesitant groups, except in regard to the perception of the risk of COVID-19. The three issues mentioned above could be due to the conflict of the government purchasing less effective vaccines, along with vaccine allocation and mixed-vaccine regimens, which have never been approved by any health organization. Thus, the effectiveness and side effects of mixed-vaccine regimens remain unclear. Complacency and confidence play a major role in convincing people who feel skeptical of or resistant to COVID-19 vaccination in Thailand. Likewise, original health policy data from a 32-country survey showed that a number of issues contribute to COVID-19 vaccine hesitancy, including that the speed at which vaccines have been developed may have resulted in relaxed regulatory standards and that there were no previously approved mRNA vaccines [6]. Therefore, individuals must acquire accurate information and be certain of vaccine safety. A solution for confidence may involve providing a highly effective vaccine that is approved by the Food and Drug Administration or a reliable health organization with known side effects. Complacency could be solved by conducting a campaign to inform the population regarding COVID-19 and the risks and benefits of vaccination and provide support for vaccination. Moreover, tax deductions or incentives may provide reinforcement.

These findings support the findings of a public health review on vaccine hesitancy, which found that women, young people, and non-healthcare workers had higher vaccine hesitancy. However, respondents with a higher level of education exhibited less hesitancy [7]. In contrast, in Sub-Saharan Africa, better-educated respondents reported substantially more reluctance, according to a study on low-and medium-income nations [33]. The reason people with higher degrees are more hesitant may be because exaggerated harmful vaccination information is widely disseminated among these people. This issue might be addressed by providing the most accurate information while limiting the propagation of unreliable information.

There is more concern in rural areas, which may reflect lower vaccine confidence compared to urban populations, as encouraging vaccination may be difficult due to geographical limitations. Similar findings have been reported in research on COVID-19 vaccine hesitancy in the U.S., which revealed that hesitancy among rural people was higher than in the general population, with 35% of rural respondents saying they would probably or definitely not receive the vaccination [34]. Therefore, the enhancement of vaccine trust in rural areas and efforts to decentralize vaccination centers may help solve this problem.

Higher-income groups demonstrated stronger hesitancy. A study on the identification and characteristics of vaccine refusers in the U.S. revealed similar results, where families who denied immunization for their children were more likely to live in neighborhoods with higher household incomes than families who did not refuse vaccination [35]. The argument is that people in middle-to-high-income areas may have easier access to information via the internet and social media, potentially exposing them to more anti-vaccine propaganda [36]. Additionally, in Thailand, government-provided vaccination is thought to have low efficacy; thus, people who can purchase other vaccines may choose high-efficacy vaccines. These findings indicated that vaccination programs should not be limited to areas based on socioeconomic status, as vaccine hesitancy affects all socioeconomic groups. However, a public health review noted that jobless people and those with lower income had a lower acceptance rate. Pogue et al. found that income had no impact on vaccination attitudes [37]. Further studies are required to determine why income has varying effects on vaccination attitudes and behaviors in various groups.

This study found that concern over vaccine-related adverse events is an important factor driving COVID-19 vaccine hesitancy. The WHO causality algorithm was developed for linking vaccination and its severe adverse events via autopsy [38,39]. As of the time of this study, less common serious adverse events have included anaphylaxis, cerebral venous thrombosis (CVT) [40,41,42,43], and thrombosis with thrombocytopenia syndrome (TTS) [44,45]

This study revealed that three-quarters of the respondents were willing to pay for the COVID-19 vaccine. Similarly, a study in Indonesia revealed that nearly 80% of the respondents had WTP for vaccines [46]. Duration (before 1 November 2021), the impact of COVID-19 on income (Yes), age group (45–64 years), and monthly income (THB 10,000–20,000 and above THB 40,000) corresponded to greater WTP. Monthly income has been reported to be an important WTP factor in Malaysia and Indonesia [14,46]. Conversely, higher education levels were not related to higher WTP in the present study, which contradicts a previous study in Indonesia [46].

The most acceptable average price for vaccines per dose was in the range of THB 501–1000 (USD 15.1–30.3), which accords with a Malaysian report (USD 30.70) [14] but was lower than studies in Indonesia (USD 57) [46] and Chile (USD 232–252) [19]. This study examined whether there was an increase in WTP based on an increased income among individuals earning more than THB 5000 per month; however, this group was not willing to pay over THB 1000 (USD 30.3). According to Statistics Thailand, the national average individual monthly income in Thailand was approximately USD 570.31 in 2020 [47].

Owing to a shortage in COVID-19 vaccine supply across the world, inadequate distribution of government-supported vaccines contributed to the slow pace of vaccination rollout in Thailand in the latter part of 2021. Respondents who were vaccinated before 1 November 2021 were more likely to pay for vaccines, and most of these groups were willing to pay more than THB 1000 per dose when compared to respondents who were vaccinated after 1 November 2021. This result could be explained by the availability of mRNA-based vaccines. Although government-supported mRNA-based vaccines (Pfizer) were imported to Thailand in mid-2021, free administration was not offered to general Thai citizens instantly due to limited supply, which was allocated based on priority, starting with healthcare workers and patients with one of seven indicated underlying diseases. Despite the adequacy of free vaccines with high efficacy, the government allowed private sectors to import alternative mRNA-based vaccines (Moderna) during the surge in COVID-19 infection. Before the alternative mRNA-based vaccine was freely registered in late November, parts of the general Thai population received paid vaccines from the private sector. It can be assumed that inadequate access to free-of-charge mRNA-based vaccines may cause public anger and perhaps drive a rising demand among Thais to request alternative vaccines.

The three major reasons why people were unwilling to pay for vaccines were inadequate vaccine distribution from the government (29%), government responsibility to pay for vaccine costs (25%), and an insufficient budget for any vaccine shot (20%). In Chile [19], most reported that the government should pay for the vaccine.

An acceptable average price for a vaccine per dose was THB 501–1000, compared to the set price of an mRNA-based vaccine in private hospitals of approximately THB 3300–3800 [48,49] for two doses (THB 1650–1900 per dose), including vaccine fees, insurance fees, and administration. Therefore, the government and stakeholders should provide expenditure for different prices, cover the total price of vaccines, or provide a variety of high-efficacy vaccines as alternatives that will give people more choices. Additionally, as the COVID-19 situation is constant and new variants of the virus have been emerging continuously, sufficient investment in local vaccine manufacturers and COVID-19 vaccine researchers may play a role in the long term to lower the prices of vaccines and develop vaccines with high efficacy, safety, and quality.

### 4.2. Strengths and Limitations

This study is among the first to examine COVID-19 vaccination hesitancy and WTP in Thailand and Southeast Asia. However, this study had some limitations. The COVID-19 infection and vaccination situation in Thailand was continuously changing between August 2021 and January 2022 due to vaccine supply and the promotion of vaccine confidence on social media or in the news. We cannot account for these confounding variables in this study. Therefore, respondents’ attitudes toward vaccine hesitancy and WTP for vaccines may be affected. Imprecise recall memory in individuals and self-reported information could also have affected this research. Although this online-based questionnaire was distributed to online platforms and public posters, the respondents in this study may not be fully representative of the general population across the country. Social media engagement was high, with a response rate of around 5%. The issue of incomplete data is a frequent occurrence in surveys, particularly online surveys. For this study, there were 17.0% and 7.8% of incomplete data in the vaccine hesitancy and WTP sections, respectively. Readers should be aware of this limitation to avoid the possibility of a misleading interpretation of the results. This study was unable to reach people without digital technology. Moreover, the KoBoToolbox does not provide an option to require respondents to answer all questions. Consequently, missing data from some populations may occur. Lastly, when we compared the 3C model (i.e., complacency, convenience, and confidence) from the Oxford COVID-19 vaccine hesitancy scale [17] with the 5C model (i.e., confidence, complacency, constraints, calculation, and collective responsibility) from unspecific vaccine hesitancy [5]. Consequently, our definition of vaccine hesitancy did not cover the topics of engagement in extensive information searching and a willingness to protect others.

### 4.3. Implications and Further Studies

This study has several implications. To improve the COVID-19 vaccination policy, addressing issues related to complacency and confidence may be one of the significant keys to reducing hesitancy toward COVID-19 vaccination during rapid campaigns. Additionally, the results indicated that the government should subsidize vaccine costs for the general population, which may increase the COVID-19 vaccine acceptance rate in the future.

Further studies should obtain a large study sample and distribute online-based questionnaires more broadly via a cooperative organization. This study was conducted over a long period with uncertain fluctuating circumstances in Thailand; thus, data collection during a brief period with a greater quantity of responses would minimize information bias. If data were collected over a longer period, the independent variable should include a longer range of time to answer a questionnaire to assess the pattern of change in hesitancy and WTP, which likely depends on the time period.

## 5. Conclusions

This study revealed that the prevalence of COVID-19 vaccine hesitancy and WTP were 10.4% and 77.2%, respectively. The reasons for vaccine hesitancy were low compliance and confidence. For vaccine hesitancy, demographic data (i.e., female, higher education, non-healthcare workers, and those who lived in rural areas) were associated with vaccine hesitancy. For WTP, economic status (i.e., high income and no impact of COVID-19 on income) were associated with WTP. The majority of WTP was in the range of THB 501–1000 ( USD 1 = THB 33) per dose. These results challenge the government to develop policies to solve vaccine hesitancy issues and set appropriate prices for alternative vaccines.

## Figures and Tables

**Table 1 vaccines-10-00627-t001:** Determinants of COVID-19 vaccine hesitancy.

	Questions
Complacency	Do you think you have a chance of contracting COVID-19? Do you agree that COVID-19 vaccination is beneficial to your community? Do you agree that COVID-19 vaccination can return life return to normal? Will you support your friends and family in getting vaccinated?
Convenience	Do you agree that COVID-19 vaccines are accessible? Do you understand the steps to access COVID-19 vaccination services? Do you agree that you have enough information about COVID-19 vaccines?
Confidence	Are you concerned about COVID-19 vaccine side-effects? Do you agree that COVID-19 vaccines prevent COVID-19 infection? Do you trust government policies to encourage COVID-19 vaccine acceptance? If you have the opportunity to receive the COVID-19 vaccine, will you accept it?

**Table 2 vaccines-10-00627-t002:** Relationship between demographic data and COVID-19 vaccine hesitancy (n = 585).

Demographic Data	Vaccine Non-Hesitancy n, (% Row)	Vaccine Hesitancy n, (%row)		
		Delayed	Denied	Total
Overall (n = 585)	524 (89.6)	57 (9.7)	4 (0.7)	61 (10.4)
Gender				
Male (n = 184)	172 (93.5)	11 (6.0)	1 (0.5)	12 (6.5)
Female (n = 399)	351 (88.0)	46 (11.5)	2 (0.5)	48 (12.0)
Age group (years)				
Median (min, max)	39 (18, 79)	25 (18, 45)	45 (45, 45)	25 (18, 45)
18–24 (n = 160)	136 (85.0)	24 (15.0)	0 (0.0)	24 (15.0)
25–44 (n = 181)	166 (92.7)	15 (8.3)	0 (0.0)	15 (8.3)
45–64 (n = 177)	161 (91.0)	14 (7.9)	2 (1.1)	16 (9.0)
≥65 (n = 27)	27 (100)	0 (0.0)	0 (0.0)	0 (0.0)
Marital status				
Single (n = 388)	352 (90.7)	35 (9.0)	1 (0.3)	36 (9.3)
Married (n = 210)	192 (91.4)	17 (8.1)	1 (0.5)	18 (8.6)
Divorced (n = 15)	13 (86.6)	1 (6.7)	1 (6.7)	2 (13.4)
Other (n = 20)	18 (90.0)	2 (10.0)	0 (0.0)	2 (10.0)
Religion				
Buddhism (n = 525)	475 (90.5)	49 (9.3)	1 (0.2)	50 (9.5)
Islamic (n = 39)	31 (79.5)	6 (15.4)	2 (5.1)	8 (20.5)
Christian (n = 9)	9 (100.0)	0 (0.0)	0 (0.0)	0 (0.0)
Other (n = 8)	6 (75.0)	2 (25.0)	0 (0.0)	2 (25.0)
Education				
Less than elementary school (n = 9)	7 (77.8)	2 (22.2)	0 (0.0)	2 (22.2)
Elementary school (n = 81)	70 (86.5)	10 (12.3)	1 (1.2)	11 (13.5)
High school / Vocational certificate (n = 35)	35 (100.0)	0 (0.0)	0 (0.0)	0 (0.0)
Bachelor’s degree or above (n = 457)	410 (89.8)	45 (9.8)	2 (0.4)	47 (10.2)
Healthcare worker				
Yes (n = 220)	197 (89.5)	23 (10.5)	0 (0.0)	23 (10.5)
No (n = 345)	308 (89.2)	34 (9.9)	3 (0.9)	37 (10.8)
Residence				
Rural (n = 74)	63 (95.1)	11 (4.9)	0 (0.0)	11 (4.9)
Suburban (n = 175)	162 (92.6)	11 (6.3)	2 (1.1)	13 (7.4)
Urban (n = 384)	348 (90.6)	35 (9.1)	1 (0.3)	36 (9.4)
Monthly income (THB)				
≤5000 (n = 81)	67 (82.7)	14 (17.3)	0 (0.0)	14 (17.3)
5001–10,000 (n = 97)	88 (90.7)	9 (9.3)	0 (0.0)	9 (9.3)
10,001–20,000 (n = 116)	104 (89.3)	12 (10.3)	0 (0.0)	12 (10.3)
20,001–40,000 (n = 126)	117 (92.9)	7 (5.6)	2 (1.6)	9 (7.1)
≥40,000 (n = 110)	103 (93.6)	6 (5.5)	1 (0.9)	7 (6.4)

Delayed: received at least two doses at least two weeks from the last dose and previously refused vaccination. Denied: refused vaccination up to the time of the study; min = minimum; max = maximum. All *p*-values < 0.05: comparison between vaccine hesitancy and non-hesitancy group.

**Table 3 vaccines-10-00627-t003:** Logistic regression model predicting COVID-19 vaccine hesitancy (n = 585).

Demographic Data	Crude OR (95%CI)	Adjusted OR (95%CI)
Gender (ref. = male)		
Female	1.05 * (1.002–1.11)	1.11 * (1.01–1.23)
Age group (years) (ref. = 18–24)		
25–44	0.92 * (0.88–0.97)	0.89 * (0.86–0.93)
45–64	1.02 (0.91–1.16)	0.95 (0.83–1.12)
≥65	0.96 (0.69–1.22)	1.00 (0.97–1.03)
Marital status (ref. = single)		
Other (married, divorced, widowed, etc.)	0.82 (0.48–1.21)	0.96 (0.87–1.05)
Religion (ref. = Buddhism)		
Islamic	0.94 (0.33–1.51)	0.72 (0.26–1.76)
Other	1.11 (0.94–1.82)	1.25 (0.67–1.75)
Education (ref. = vocational certificate or below)		
Bachelor’s degree or above	1.43 * (1.22–1.78)	1.28 * (1.04–1.57)
Healthcare (ref. = healthcare worker)		
Non-healthcare worker	1.11 * (1.02–1.20)	1.23 * (1.17–1.29)
Residence (ref. = urban)		
Rural	1.07 * (1.01–1.13)	1.22 * (1.09–1.35)
Suburban	1.02 (0.98–1.07)	1.01 (0.94–1.07)
Monthly income (THB) (ref. = ≤5000)		
5001–10,000	1.03 (0.96–1.11)	0.99 (0.92–1.08)
10,001–20,000	0.88 (0.72–1.07)	0.86 (0.67–1.01)
20,001–40,000	1.17 * (1.01–1.33)	1.15 * (1.10–1.21)
≥40,000	0.99 (0.84–1.27)	1.07 (0.97–1.34)

OR = odds ratio, CI: confidence interval, * *p*-value < 0.05.

**Table 4 vaccines-10-00627-t004:** COVID-19 vaccine hesitancy (n = 585).

Issue	Score from 1 to 5, Median (IQR)		*p*-Value
	Hesitancy (n = 61)	Non-Hesitancy (n = 524)	
Complacency			
Risk of COVID-19 infection	3.8 (3.0, 4.0)	3.9 (3.0, 5.0)	0.460
Beneficence of vaccine	4.2 (4.0, 5.0)	4.5 (4.0, 5.0)	0.009
Daily life impact	3.3 (3.0, 4.0)	3.7 (3.0, 5.0)	0.001
Support for vaccination	4.1 (4.0, 5.0)	4.5 (4.0, 5.0)	<0.001
Convenience			
Vaccine accessibility	3.3 (2.5, 4.0)	3.4 (3.0, 4.0)	0.324
Process comprehension	4.1 (4.0, 4.5)	4.1 (4.0, 5.0)	0.285
Vaccine comprehension	3.7 (3.0, 4.0)	3.9 (3.0, 4.0)	0.402
Confident			
Lack of concern regarding side effects	2.7 (2.0, 3.0)	3.2 (2.0, 4.0)	<0.001
Trust in effectiveness	2.6 (2.0, 3.0)	3.2 (2.8, 4.0)	<0.001
Trust in vaccine policies	2.6 (1.0, 3.5)	3.1 (2.0, 4.0)	<0.001
Vaccine necessity	3.8 (3.0, 5.0)	4.4 (4.0, 5.0)	<0.001

IQR: interquartile range. *p*-value: compared between hesitancy and non-hesitancy group, using Mann–Whitney U test.

**Table 5 vaccines-10-00627-t005:** Relationship between demographic characteristics, vaccine hesitancy, and percentage of willingness to pay for the COVID-19 vaccine (n = 650).

Demographic Data	Willingness to Pay n (%)	*p*-Value
Overall (n = 650)	502 (77.2)	
Gender		
Male (n = 206)	156 (75.3)	0.098
Female (n = 443)	345 (78.9)	
Marital status		
Single (n = 381)	290 (76.1)	0.039
Married (n = 232)	183 (78.9)	
Other (married, divorced, widowed, etc.) (n = 37)	28 (75.7)	
Age group (years)		
18–24 (n = 180)	126 (70)	0.036
25–44 (n = 215)	176 (81.9)	
45–64 (n = 188)	151 (80.3)	
≥65 (n = 29)	21 (72.4)	
Education		
Less than high school (n = 147)	107 (72.8)	<0.001
Vocational certificate or above (n = 498)	390 (78.3)	
Monthly income (THB)		
≤5000 (n = 78)	61 (78.2)	0.015
5001–10,000 (n = 114)	97 (85.1)	
10,001–20,000 (n = 133)	112 (84.2)	
20,001–40,000 (n = 139)	113 (81.3)	
≥40,000 (n = 112)	112 (92.6)	
Impact of COVID-19 on income		
Yes (n = 380)	279 (73.4%)	0.002
No (n = 232)	190 (81.9%)	
Residence		
Urban (n = 425)	321 (75.6)	0.076
Suburban (n = 139)	114 (82.0)	
Rural (n = 85)	66 (77.6)	
Duration		
Before 1 November 2021 (n = 159)	138 (86.8)	<0.001
Since 1 November 2021 (n = 491)	364 (74.1)	
Vaccine hesitancy		
Hesitancy (n = 61)	51 (83.6%)	<0.001
Non-hesitancy (n = 461)	395 (85.7%)	

USD 1 = THB 33. *p*-value: comparison between the willingness to pay and non-willingness to pay groups.

**Table 6 vaccines-10-00627-t006:** Logistic regression model predicting willingness to pay for the COVID-19 vaccine (n = 650).

Demographic Data	Crude OR (95%CI)	Adjusted OR (95%CI)
Gender (ref. = male)		
Female	1.01 (0.53–1.91)	1.02 (0.95–1.09)
Marital status (ref. = single)		
Married	2.19 (0.72–3.21)	1.42 (0.82–2.49)
Other (married, divorced, widowed, etc.)	1.79 (0.92–2.17)	1.31 (1.07–2.14)
Age group (years) (ref. = 18–24)		
25–44	1.55 (0.83–2.42)	1.44 (0.63–2.31)
45–64	1.48 * (1.34–1.98)	1.53 * (1.42–1.88)
≥65	1.64 * (1.33–1.77)	1.31 (0.91–1.79)
Education (ref. = less than high school)		
Vocational certificate or above	1.07 (0.89–1.98)	1.05 (0.97–1.14)
Monthly income (THB) (ref. = ≤5,000)		
5001–10,000	0.94 (0.82–1.02)	0.87 (0.80–1.01)
10,001–20,000	1.29 * (1.11–1.89)	1.05 * (1.02–1.07)
20,001–40,000	1.03 (0.99–1.24)	1.08 (0.94–1.53)
≥40,000	1.88 * (1.67–2.13)	1.67 * (1.44–1.93)
Impact of COVID-19 on income (ref. = yes)		
No	1.66 * (1.34–2.05)	1.31 * (1.06–1.68)
Residence (ref. = urban)		
Suburban	0.89 (0.74–1.22)	1.02 (0.98–1.10)
Rural	1.01 (0.79–1.04)	1.07 (0.82–1.19)
Duration (ref. = before 1 November 2021)		
Since 1 November 2021	0.77 * (0.52–0.91)	0.90 * (0.84–0.97)
Vaccine hesitancy (ref. = non-hesitancy)		
Hesitancy	1.15 * (1.11–1.22)	1.01 (0.99–1.04)

OR: odds ratio, CI: confidence interval, USD 1 = THB 33, * *p*-value < 0.05.

**Table 7 vaccines-10-00627-t007:** Relationship between the range of willingness to pay for COVID-19 vaccine and demographic data and vaccine hesitancy (n = 502).

Factor	Range of Willingness to Pay in THB, Row%						
	125–250	251–500	501–1000	1001–2000	2001–4000	4001–8000	>8000
Overall (n = 502)	11.6	19.9	27.3	23.9	13.7	1.8	1.8
Gender							
Male (n = 156)	10.9	14.1	28.8	27.6	12.2	2.6	1.9
Female (n = 345)	11.6	22.6	26.7	22.3	14.5	1.4	0.0
Marital status							
Single (n = 290)	9.7	17.9	24.8	26.6	15.9	2.8	2.4
Married (n = 183)	14.2	24.6	30.1	20.8	10.4	0.5	0.5
Other (divorced, widowed, etc.) (n = 28)	10.7	17.9	35.7	17.9	14.3	0.0	3.6
Age group (years)							
18–24 (n = 126)	13.5	12.7	27.8	24.6	16.7	2.4	2.4
25–44 (n = 176)	6.3	21.0	23.3	30.1	15.3	1.7	2.3
45–64 (n = 128)	13.3	27.3	28.1	20.3	8.6	0.8	1.6
≥65 (n = 43)	18.6	20.9	39.5	14.0	7.0	0.0	0.0
Education							
Less than high school (n = 107)	22.4	22.4	22.4	15.9	13.1	4.7	0.0
Vocational certificate or above (n = 390)	8.5	19.5	28.5	26.4	14.1	0.8	2.3
Monthly income (THB)							
≤5000 (n = 61)	59.0	8.2	4.9	21.3	6.6	0.0	0.0
5001–10,000 (n = 97)	25.8	13.4	44.3	7.2	3.1	0.0	0.0
10,001–20,000 (n = 122)	9.0	23.0	27.0	22.1	13.1	4.1	1.6
20,001–40,000 (n = 113)	7.1	21.2	31.9	24.8	11.5	1.8	1.8
≥40,000 (n = 112)	6.3	15.2	30.4	25.0	17.0	3.6	2.7
Impact of COVID-19 on income							
Yes (n = 279)	15.1	22.6	28.0	20.8	11.8	1.4	0.4
No (n = 190)	11.1	18.4	28.9	28.4	13.7	1.6	2.6
Residence							
Urban (n = 321)	9.7	19.3	28.0	24.3	14.6	1.6	2.5
Suburban (n = 114)	13.2	20.2	31.6	23.7	10.5	0.0	0.9
Rural (n = 66)	16.7	22.7	16.7	22.7	15.2	6.1	0.0
Duration							
Before 1 November 2021 (n = 138)	7.2	8.7	25.4	27.5	23.2	2.9	5.1
Since 1 November 2021 (n = 364)	13.2	24.2	28.0	22.5	10.2	1.4	0.5
Vaccine hesitancy							
Hesitancy (n = 51)	19.6	15.7	15.7	25.5	19.6	2.0	2.0
Non-hesitancy (n = 395)	10.9	21.5	26.8	23.0	14.7	1.5	1.5

USD 1 = THB 33.

## Data Availability

Not applicable.

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
