# Peer review of "Self-Reported COVID-19 Vaccine Hesitancy and Willingness to Pay: A Cross-Sectional Survey in Thailand"

_vaccines, 2022, doi:10.3390/vaccines10040627_

Round 1

Reviewer 1 Report

The paper is written in a clear and simple way, however there are some points that should be revised:
1) The abstract and introduction are well argued and clear;
2) Line 75 - what is meant by "public poster"? Authors should make this clear;
3) The materials and methods, the results are clearly presented, however the age range (minimum age-maximum age and median age) must be clarified.
4) In the discussion the authors should make a comparison with the different nations, these studies should be cited:
10.3390 / vaccines9101200
https://doi.org/10.1016/j.eclinm.2021.101113
10.3390 / vaccines9121415

Finally, the concern about the side effects of vaccines also needs to be clarified. Authors should make a small paragraph on the subject. These studies should be cited:
10.3390 / healthcare10020319
10.3389 / fneur.2021.721146
10.1002 / ajh.26336
10.3390 / diagnostics11060955
10.3346 / JKMS.2021.36.E223
10.1016 / j.bbi.2021.04.006
10.3390 / ijerph19052721

Author Response

Thank you very much for your valuable comments and suggestions.

Response to Reviewers

Manuscript ID: vaccines-1639115

Manuscript title: Self-reported COVID-19 vaccine hesitancy and willingness to pay: A cross-sectional survey in Thailand

Reviewer #1: 

The paper is written in a clear and simple way, however there are some points that should be revised:

1) The abstract and introduction are well argued and clear;

2) Line 75 - what is meant by "public poster"? Authors should make this clear;

Response:  

Revised accordingly. We have revised the entire sentence to make it clearer. It now reads: We also placed the posters with QR code for recruiting the participants in public places.” (Line 7576)

3) The materials and methods, the results are clearly presented, however the age range (minimum age-maximum age and median age) must be clarified.

Response:  Revised accordingly in Table 2.

4) In the discussion the authors should make a comparison with the different nations, these studies should be cited:
10.3390 / vaccines9101200
https://doi.org/10.1016/j.eclinm.2021.101113
10.3390 / vaccines9121415

Response:

Following the reviewer’s suggestion, the reference has been added to our revised manuscript. (Line 268–271)

Finally, the concern about the side effects of vaccines also needs to be clarified. Authors should make a small paragraph on the subject. These studies should be cited:
10.3390 / healthcare10020319
10.3389 / fneur.2021.721146
10.1002 / ajh.26336
10.3390 / diagnostics11060955
10.3346 / JKMS.2021.36.E223
10.1016 / j.bbi.2021.04.006
10.3390 / ijerph19052721

Response:  

Thank you verry much for the reviewer’s advice. In the revised version, we have added a paragraph describing the vaccine adverse events to the revised manuscript. (Line 578–583)

Reviewer 2 Report

I was invited to revise the paper entitled "Self-reported COVID-19 vaccine hesitancy and willingness to pay: A cross-sectional survey in Thailand". It was a cross-sectional study aiming to investigate factors associated to vaccine hesitancy and willingness to pay among inhabitants from a middle income countries. In my knowledge this is one of the first paper that investigated the WTP towards covid vaccine. The topic is interesting but I have some observations:

  • Sample size estimation technique should be better described;
  • Validation tecnique of the survey was not reported;
  • Table 2 should report also non-hesitant patients characteristics. In addition they should be considered for the analysis;
  • Table 3 should report the frequency of each variables. In addition, it is unclear if non-hesitant patients were considered;
  • Authors should report also univariate analyses results in table 3 and 6. In addition authors shoulds better describe the type of multivariate logistic models: full adjusted model? backward/forward selection?
  • Variables were tested for multicollinearity?
  • It is unclear why the number of patients enrolled for the WTP survey was greater then VH survey. Authors should better explain enrollment procedure and methods;
  • Age category were not euqual. Why? This is not justified in my opinion. In addition patients over 80 can have more WTP compared to 60-70 group due to more frail health status;
  • Table 7 should be better described in methods and results;

Author Response

Thank you very much for your valuable comments and suggestions.

Response to Reviewers

Manuscript ID: vaccines-1639115

Manuscript title: Self-reported COVID-19 vaccine hesitancy and willingness to pay: A cross-sectional survey in Thailand

Reviewer #2: 

I was invited to revise the paper entitled "Self-reported COVID-19 vaccine hesitancy and willingness to pay: A cross-sectional survey in Thailand". It was a cross-sectional study aiming to investigate factors associated to vaccine hesitancy and willingness to pay among inhabitants from a middle income countries. In my knowledge this is one of the first paper that investigated the WTP towards covid vaccine. The topic is interesting but I have some observations:

Sample size estimation technique should be better described;

Response: Revised accordingly.

Validation tecnique of the survey was not reported;

Response:

In this study, we used the standard questionnaire from Oxford and WHO. Moreover, we applied the face validity of three content experts, and also conducted the pilot study. These were mentioned as below;

The questionnaire was developed based on the Oxford COVID-19 vaccine hesitancy scale [15], definition by the WHO [8], and WTP scale [16,17]. Each item was validated using face validity by three content experts before conducting a pilot study with 30 respondents to ensure a similar understanding.”  (Line 8790)

Table 2 should report also non-hesitant patients characteristics. In addition they should be considered for the analysis;

Response:

Revised accordingly. For Table 2, we added a new column presenting non-hesitancy group.

Table 3 should report the frequency of each variables. In addition, it is unclear if non-hesitant patients were considered;

Response: Revised accordingly (Table 3).

Authors should report also univariate analyses results in table 3 and 6. In addition authors shoulds better describe the type of multivariate logistic models: full adjusted model? backward/forward selection?

Response:

Revised accordingly. In revised version, we added crude odds ratio (OR) for univariate analysis in Table 3 and 6.

For the final model in multivariate logistic regression, we provided more details on model adjustment using a manual backward stepwise refinement. This information has been added to the “Statistical analysis” section (Line 161–163).

Variables were tested for multicollinearity?

Response: Revised accordingly (statistical analysis part: Line 163–165).

The refined model had to pass the multicollinearity assumption. A variance inflation factor (VIF) greater than five was indicated the collinearity between our independent variables. “  

It is unclear why the number of patients enrolled for the WTP survey was greater then VH survey. Authors should better explain enrollment procedure and methods;

Response:

We would like to thank the reviewer for pointing out the confusion. There were more participants in WTP, compared with vaccine hesitancy because “those who have not been offered any vaccines” recruited in WTP, not vaccine hesitancy.

In the revised manuscript, we added a new sentence at the first paragraph of “Result” section (Line 167-170): Among the 705 respondents, 585 (83.0%) participants (i.e., vaccinated, delayed vaccinated, and denied vaccinated) provided answers regarding vaccine hesitancy. For WTP, there were 650 (92.2%) participants provided answers (i.e., vaccinated, delayed vaccinated, denied vaccinated, and those who have not been offered any vaccines).

For the “Study sample” section (Line 78-80), the paragraph has been rewritten and now reads: “The respondents were vaccinated (i.e., suddenly vaccinated and delayed vaccinated) and unvaccinated (i.e., denied vaccinated and those who have not been offered any vaccines) Thai people aged ≥18 years who owned smartphones or could access internet services.”

Age category were not euqual. Why? This is not justified in my opinion. In addition patients over 80 can have more WTP compared to 60-70 group due to more frail health status;

Response:

Regarding with COVID-19 vaccine, there are no standard cutoffs for age group. In this study, we categorized the age group by balancing the number of participants in each age group.

We agreed with reviewer’s comment on the elderly people issue. However, this was not our primary objective and there were only 29 participants aged 65 and older in this study. Thus, we have decided to apply the cut-off point at 65 years.

Table 7 should be better described in methods and results;

Response:

Revised accordingly. We added an explanation of what a heatmap (or highlight table) in the “Statistical analysis” section: Heatmap or highlight table was visualized the relationship between WTP and demographic data and vaccine hesitancy. The color scale was green to yellow and red with high percentage getting the green color and low percentage getting the red color.” (Line 156159)

Reviewer 3 Report

Mueangpoon and colleagues studied the he prevalence and influencing factors of COVID-19 vaccine hesitancy and willingness to pay in Thailand.  The authors showed that  10.4% of responders reported hesitancy, with significant determinants being low complacency and confidence in the vaccine.  Women, adults (24–44 years), those with higher education, non-healthcare workers, and those who lived in rural areas are related to the vaccine hesitancy.

The study is important, I have some suggestions

a) Please include mortality rate in Thailand associated with COVID-19 infection, in those who are vaccinated, non-vaccinated, or partial vaccinated. Please mention the demographic criteria for those people.

b) Please include COVID-19 complication in Thailand associated with  in those who are non-vaccinated, or partial vaccinated. Please mention the demographic criteria for those people.

c) Does the social media and news have effect on vaccine hesitancy in Thailand?

Author Response

Thank you very much for your valuable comments and suggestions.

Response to Reviewers

Manuscript ID: vaccines-1639115

Manuscript title: Self-reported COVID-19 vaccine hesitancy and willingness to pay: A cross-sectional survey in Thailand

Reviewer #3: 

Mueangpoon and colleagues studied the he prevalence and influencing factors of COVID-19 vaccine hesitancy and willingness to pay in Thailand.  The authors showed that  10.4% of responders reported hesitancy, with significant determinants being low complacency and confidence in the vaccine.  Women, adults (24–44 years), those with higher education, non-healthcare workers, and those who lived in rural areas are related to the vaccine hesitancy.

The study is important, I have some suggestions

  1. a) Please include mortality rate in Thailand associated with COVID-19 infection, in those who are vaccinated, non-vaccinated, or partial vaccinated. Please mention the demographic criteria for those people.

Response:

We agree with the reviewer that the COVID-19 specific mortality rate is one of most important indicators. We have asked the mortality rate from Department of Disease Control, Ministry of Public Health, Thailand. The Department of Disease Control answered that the report or peer-review research were limited due to invalidated data.

By the way, we have addressed this topic in relation to studies in England and Israel: “Additionally, recent evidence has shown that fully vaccinated people have a lower incidence rate of COVID-19 infection, COVID-19-related hospitalization, severe symptoms, and a lower mortality rate than unvaccinated people [6,7]." (Line 3133)

  1. b) Please include COVID-19 complication in Thailand associated with  in those who are non-vaccinated, or partial vaccinated. Please mention the demographic criteria for those people.

Response:

We agree with the reviewer’s comment that the complications and severity of COVID-19 are vital indices. Similar to prior reviewer’s comment, we have asked the mortality rate from Department of Disease Control, Ministry of Public Health, Thailand. The Department of Disease Control answered that the report or peer-review research were limited due to invalidated data.

  1. c) Does the social media and news have effect on vaccine hesitancy in Thailand?

Response: We agree with the reviewer and truly appreciate the comment. However, these variables (i.e., effects of social media/news on vaccine hesitancy) cannot collect in this study. Thus, we added this issue in the “Limitation” section that these confounders can affect our outcomes.

“The COVID-19 infection and vaccination situation in Thailand was continuously changing between August 2021 and January 2022 due to vaccine supply and promote vaccine confidence on social media or news. We cannot collect these confounding variables in this study. Therefore, respondents’ attitudes toward vaccine hesitancy and WTP for vaccines may be affected.” (Line 456460)

Round 2

Reviewer 1 Report

The manuscript has been sufficiently improved, however there are some minor corrections:
delete lines 27-35 because they are useless for the purposes of the study.

Author Response

Revised accordingly, we would like to thank the reviewer for pointing out this. 

Reviewer 2 Report

I was invited to review the revised version of the paper entitled "Self-reported COVID-19 vaccine hesitancy and willingness to pay: A cross-sectional survey in Thailand". 

Despite some improvements, I have some concerns:

  • About sample size estimation, Why Authors choosed 0.33 as proportion of hesitant subject? References? Margin error 0.04?? It seems to be post estimatated; In addition, wich is the population sampled?
  • Validation and internal consistency of the questionnarire is still missing;
  • Line 153 - Logistic regressionis not a prediction model!!!
  • Statistical analysis explanation should be better described;
  • Age categories should be equal and not based on the distribution of enrolled patients;
  • Authors did not described how they handled missing data, as reported in limitation section.

Author Response

Thank you very much. We appreciate your valuable feedback.

was invited to review the revised version of the paper entitled "Self-reported COVID-19 vaccine hesitancy and willingness to pay: A cross-sectional survey in Thailand". 

Despite some improvements, I have some concerns:

  1. About sample size estimation, Why Authors choosed 0.33 as proportion of hesitant subject? References? Margin error 0.04?? It seems to be post estimatated; In addition, wich is the population sampled?

Response:

We would like to thank the reviewer for pointing out the confusion. However, we calculated the sample size before start data collection.

The researchers accepted the margin error = 0.04. This number was calculated from about 10% to 15% of our proportion of vaccine hesitancy: 10%-15% of 0.33 = 0.033–0.0495 (mean of this = 0.04). (Ref: Sample size determination in health studies : a practical manual / S. K. Lwanga and S. Lemeshow (who.int))

We selected the study in Japanese population, whose race/culture/others are familiar with Thai culture.

  1. Validation and internal consistency of the questionnarire is still missing;

Response:

      Thank you for the reviewer’s suggestion. We added the validity and reliability (i.e., internal consistency) in the revised manuscript.

It now reads: “The scores of the Oxford COVID-19 Vaccine Hesitancy Scale scores were correlated with scores of the Vaccine Hesitancy Scale (correlation coefficient = 0.47, p<0.001) [18]. The Cronbach’s alpha was 0.97 [15].”

  1. Line 153 - Logistic regressionis not a prediction model!!!

Response:

We agree with the reviewer’s comment. We have changed the word “potential predictors” to “potential independent variables”.

  1. Statistical analysis explanation should be better described;

Response:

      Revised accordingly. We have revised a whole “Statistical analysis” section. In brief, we added the details of (i) data management, (ii) complete-case analysis for missing data, (iii) descriptive statistics, (iv) statistics test for categorical and continuous data, (v) data visualization via heat-map, and (vi) regression analysis.

  1. Age categories should be equal and not based on the distribution of enrolled patients;

Response:

We would like to thank the reviewer for pointing out the interesting issue. We have reviewed the standard cut-off points for age. We found that

The United Nations (U.N) set out provisional guidelines on standard international age classifications (Ref: https://unstats.un.org/unsd/publication/SeriesM/SeriesM_74e.pdf). Based on existing national practices and international recommendations, the U.N recommended the following:

  • The development of international age classifications into 12 different subject areas available at three different levels of detail:
    • The lowest level consists of six broad population groups -comparable to infancy, youth, young adulthood, middle adulthood, older adulthood to average retirement and retirement (under 1, 1-14, 15-24, 25-44, 45-64 and 65 and over).
    • The medium level uses combinations of five and ten year groupings
    • The highest level uses combinations of single and five year groupings ending on four and nine
  1. Authors did not described how they handled missing data, as reported in limitation section.

Response:

         We agree with the reviewer’s comment. We added information on a complete–case analysis (not impute) for dealing with missing data in the “Statistical analysis” section. We also added the missing data issue on our “Limitation” section.

Reviewer 3 Report

The authors addressed my questions

Author Response

Thank you very much. We appreciate your valuable feedback.

Round 3

Reviewer 2 Report

Authors responded to all points.

I want to highlight that logistic models analyze the association between the indipendednt and the dependent variable.

Author Response

Revised accordingly.